# Symbionts protect aphids from parasitic wasps by attenuating herbivore-induced plant volatiles

Enric Frago [1,2], Mukta Mala[1], Berhane T. Weldegergis[1], Chenjiao Yang[1], Ailsa McLean[3], H.Charles J. Godfray[3], Rieta Gols [1] & Marcel Dicke [1]

Plants respond to insect attack by releasing blends of volatile chemicals that attract their herbivores' specific natural enemies, while insect herbivores may carry endosymbiotic microorganisms that directly improve herbivore survival after natural enemy attack. Here we demonstrate that the two phenomena can be linked. Plants fed upon by pea aphids release volatiles that attract parasitic wasps, and the pea aphid can carry facultative endosymbiotic bacteria that prevent the development of the parasitic wasp larva and thus markedly improve aphid survival after wasp attack. We show that these endosymbionts also attenuate the systemic release of volatiles by plants after aphid attack, reducing parasitic wasp recruitment and increasing aphid fitness. Our results reveal a novel mechanism through which symbionts can benefit their hosts and emphasise the importance of considering the microbiome in understanding insect ecological interactions.

[1] Laboratory of Entomology, Wageningen University, P.O. Box 16,6700AA Wageningen, The Netherlands. [2] CIRAD, UMR PVBMT, Saint-Pierre, La Réunion F-97410, France. [3] Department of Zoology, University of Oxford, South Parks Road, Oxford, OX1 3PS, UK. Correspondence and requests for materials should be addressed to E.F. (email: enric.frago@cirad.fr)

Associations between microbial symbionts and multicellular eukaryotes are widespread in nature and many organisms rely on symbionts for a variety of functions important for their survival and reproduction[1]. Acquisition of symbionts can be key innovations that allow diversification into unexploited adaptive zones. Many insects depend on bacterial symbionts to provide essential nutrients that are otherwise missing from their diets[2]. These types of symbionts are termed obligate as they are essential for survival. There is increasing interest in the role played by facultative symbionts that, while not essential for survival or reproduction, provide important services for their hosts. In the last 20 years, the discovery that many facultative symbionts help protect their hosts from natural enemies has transformed our understanding of how insect symbionts affect the interactions between their hosts and higher trophic levels[3]. Symbiont-conferred protection against pathogens, parasitic wasps and predators has been demonstrated in a variety of different species, and recent evidence suggests that they can also help herbivores overcome specific induced defences mounted by plants in response to insect attack[4]. For example, the whitefly *Bemisia tabaci* and the Colorado potato beetle *Leptinotarsa decemlineata* carry facultative symbionts that manipulate host plant physiology through salivary effectors that attenuate induced defences to the benefit of their hosts[5, 6].

Natural enemies of herbivorous insects commonly use volatile chemical cues to locate their often concealed hosts or prey in the structurally complex environment which they inhabit[7]. Mutualistic symbionts could affect the likelihood of their hosts' discovery in two ways. First, they might produce "infochemicals" that attract natural enemies. For example, bark beetles carry symbiotic fungi which they use to digest wood, but the symbiont also releases volatiles that attract parasitic wasps[8, 9]. Here the nutritional benefit provided by the symbiont may be counteracted by increased attraction to natural enemies, something that may lead to the loss of the microbial partner in the host population when natural enemy pressure is high. Second, symbionts may interfere with the plant's ability to attract its herbivore's natural enemies so benefiting the host. Plants often respond to herbivore attack by releasing a specific blend of volatiles that attract the insect's natural enemies (so-called 'bodyguard recruitment')[10]. We do not know whether the presence of facultative symbionts interferes with the induction of this indirect herbivore-defence mechanism.

Here we studied symbiont-herbivore-natural enemy interactions on the pea aphid (*Acyrthosiphon pisum*) feeding on the broad bean *Vicia faba*. This plant is known to respond to pea aphid attack by releasing volatiles that attract the parasitic wasp *Aphidius ervi*, and olfactometer and biochemical studies have identified the specific volatiles that are involved in wasp attraction[11–14]. These studies have also shown that parasitic wasps are attracted to blends of volatiles rather than to individual compounds[15], yet the effective blend has not yet been elucidated for any parasitoid species. Pea aphids are associated with at least eight different facultative symbionts[16–18] including *Hamiltonella defensa*, many strains of which increase survival after parasitic wasp attack[19, 20]. Using olfactometer choice experiments, we investigated whether pea aphids benefit from carrying the symbiont *H. defensa* by influencing the release of volatiles and reducing parasitic wasp recruitment, and whether any effect on volatile release was localised to the site of aphid attack or systemic. There are costs to carrying the symbiont[21–23] and in further experiments we assessed for any effect of aphid vigour on the plant's response. To do this, we measured aphid offspring (a good measure of vigour), and performed experiments infesting bean plants with a varying number of aphids. Different populations (or biotypes) of pea aphid are adapted to different host plants, though

all feed on *V. faba*, which is considered a "universal" host[24]. Most of our experiments were performed on *V. faba* and we asked whether the natural host plants (*Ononis spinosa* and *Lotus pedunculatus*) of two different biotypes originally collected on these plants showed the same response. We carried out experiments in microcosm cages to test whether differential recruitment of parasitic wasps translated into reduced parasitism. The blend of volatiles present in the 'headspace' around plants fed upon by symbiont-carrying and uninfected aphids was characterised to identify the mechanistic basis of the effect we observed.

Finally, we carried out a more limited set of olfactometer experiments to test whether carriage of four other endosymbionts affected recruitment of the parasitic wasp *A. ervi*. The species we studied were (i) *Regiella insecticola*, which typically protects *A. pisum* from specialist pathogenic fungi but not parasitic wasps[25]; (ii) *Spiroplasma* sp. which, depending on the particular isolate, may or may not confer protection against wasps[26, 27]; (iii) *Serratia symbiotica*, which also shows some strain-specific parasitic wasp protection[20] and protects aphids from heat shocks[28]; and (iv) *Rickettsiella* sp., which is less well characterised but is known to influence aphid body colour and hence possibly attraction to natural enemies[29].

Our study shows that plants infested with aphids carrying different symbiont species and strains are less attractive to the parasitic wasp *A. ervi* through systemic changes in herbivore-induced plant volatiles, ultimately reducing parasitic wasp recruitment and increasing aphid fitness. We demonstrate this with behavioural experiments, but also analysing volatile blends. Relative to plants fed upon by symbiont-free aphids, blends of plants fed upon by aphids carrying the symbiont *H. defensa* are different and total emissions are lower. Our results reveal a novel mechanism by which insect symbionts protect their hosts through manipulation of induced defences mounted by plants in response to insect attack.

## Results

**Effect of symbiont *H. defensa* on parasitic wasp attraction.** Across a large set of experiments with different symbiont strains and plant species, and using choice experiments, parasitic wasps were significantly less likely to be attracted to plants that had previously been fed on by aphids carrying *H. defensa* compared to plants fed on by aphids that carried no secondary symbionts (Fig. 1, with statistical results in the legend). This was true in experiments where the comparison was between natural aphid-symbiont associations and the same aphid clone from which the symbiont had been removed using antibiotics (Fig. 1a, columns i–iii), and in experiments where naturally secondary symbiont-free aphids were compared to aphids which had received the symbiont through microinjection (Fig. 1a, columns iv–v). The preference was not affected by wrapping the leaf on which the aphids had fed in aluminium foil (Fig. 1a, column vi), which excludes the possibility that parasitoids are attracted to any chemical deposited by the aphid and shows that the attractant volatiles are produced systemically by the plant. The parasitoids still showed a preference for plants that had been fed on by aphids not carrying *H. defensa* when the densities of aphids with the symbiont were doubled (Fig. 1b). This means that the reduced attraction is not due to diminished damage caused by aphids carrying a potentially costly symbiont, a conclusion confirmed by the lack of a correlation between wasp preference and aphid vigour as estimated by relative progeny production (Fig. 2a, Supplementary Table 1). Two of the natural symbiont-aphid associations involved aphids belonging to biotypes associated with *Ononis spinosa* and *Lotus pedunculatus* and which had been collected on these host plants. We found the same wasp

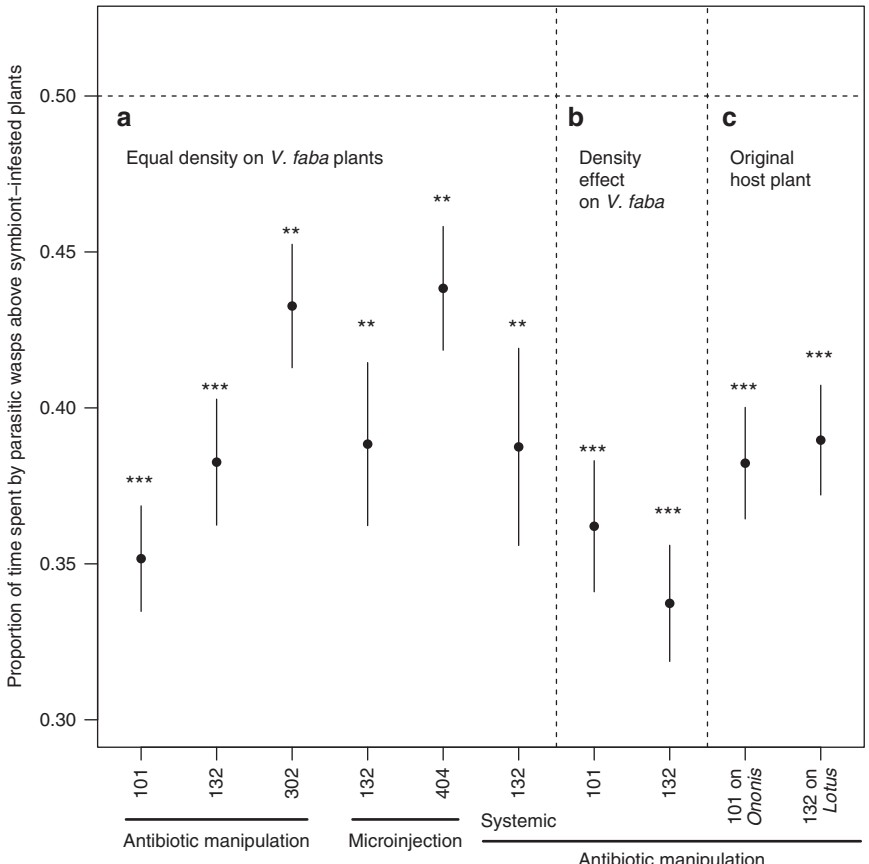

**Fig. 1** Parasitic wasp preference for volatiles from Fabaceae plants infested with aphids with or without the endosymbiont. Parasitic wasp (*Aphidius ervi*) response to volatiles emitted by plants infested with aphids (*Acyrthosiphon pisum*) with or without the endosymbiont *Hamiltonella defensa*. For each test, the bars show standard errors and the asterisks the significance of the deviation from no choice (*t*-test: \*\**p* < 0.01; \*\*\**p* < 0.001). **a** Tests where *Vicia faba* plants had been induced by 20 wingless aphid females each (*n* = 10; column i—cured 101: $t_9 = -8.78$, *p* < 0.0001; ii—cured 132: $t_9 = -5.82$, *p* = 0.0001; iii— cured 302: $t_9 = -3.41$, *p* = 0.0039, iv—microinjected 132: $t_9 = -4.27$, *p* = 0.0010; v—microinjected 404: $t_9 = -3.11$, *p* = 0.0062, vi—cured 132: $t_9 = -3.56$, *p* = 0.0031). **b** Tests with *V. faba* plants induced with double the number of symbiont-bearing compared to symbiont-free aphids (*n* = 10; column vii—cured 101: $t_9 = -6.57$, *p* < 0.0001; viii—cured 132: $t_9 = -8.73$, *p* < 0.0001). **c** Tests with the original host plants *Ononis spinosa* (*n* = 10; column ix—cured 101, $t_9 = -6.58$, *p* < 0.0001) and *Lotus pedunculatus* (column x—cured 132, $t_9 = -6.28$, *p* < 0.0001). Microinjections were performed into an aphid clone naturally lacking any secondary symbiont, and which was collected on *Lathyrus pratensis*. Note that the same symbiont strain and aphid clone may have been used in different tests

preferences when the experiments with these clones were repeated on the natural host plant as we had when working with *Vicia faba* (Fig. 1c).

**Effect of symbiont *H. defensa* on parasitic wasp attack**. We used population cage experiments to explore whether the reduced attractiveness to parasitic wasps of plants fed upon by aphids carrying *H. defensa* translates into lower rates of parasitism. Parasitism was assessed by placing 'sentinel aphids' belonging to two secondary symbiont free clones that can be recognised by a colour polymorphism on the plants immediately before the introduction of parasitoids. This polymorphism had no effect on wasp parasitism (0.80 ± 0.05 [SE] vs. 0.75 ± 0.06; paired *t*-test: $t_{14}$ = 1.26, *n* = 15, *p* = 0.2298). Aphids on plants previously infested by symbiont-carrying aphids were parasitized significantly less often than the alternative (0.62 ± 0.04 [SE] vs. 0.94 ± 0.01; paired *t*-test: $t_{14}$ = 7.73, *n* = 15, *p* < 0.0001).

**Effect of symbiont *H. defensa* on plant volatile emission**. Volatile compounds were collected from plants previously attacked by aphids carrying or not carrying *H. defensa* and analysed using gas chromatography mass spectrometry. Overall we

found 66 volatile compounds (Supplementary Table 2) and a Principal Least Squares Discriminant Analysis (PLS-DA) showed a significant difference in the volatile composition of the headspace of the two types of plant (Fig. 3a; NMC = 0.1778, *n* = 9, *p* = 0.0151). As revealed by VIP scores (variable importance in projection), treatment separation was chiefly due to 24 compounds. Among these compounds, nine were significantly more abundant in the treatment with no *H. defensa*, while we did not find any compound to be significantly more abundant in the symbiont treatment (Figs. 3b, 4, Supplementary Table 2). Of the 66 volatile compounds, 55 had mean concentrations that were greater in plants with symbiont-free aphids than in those where the symbiont was present. Overall, total emissions were significantly lower in plants fed upon by aphids carrying the symbiont compared with symbiont-free insects (sign test S = 55, *n* = 66, *p* < 0.0001).

**The effect of other symbionts on parasitic wasp recruitment**. We explored the effect of feeding by aphids carrying other facultative symbionts on wasp preferences in choice experiments. Four experiments were carried out with *Regiella insecticola*, two using natural associations and two in which different symbiont strains were introduced into an aphid clone that carried no

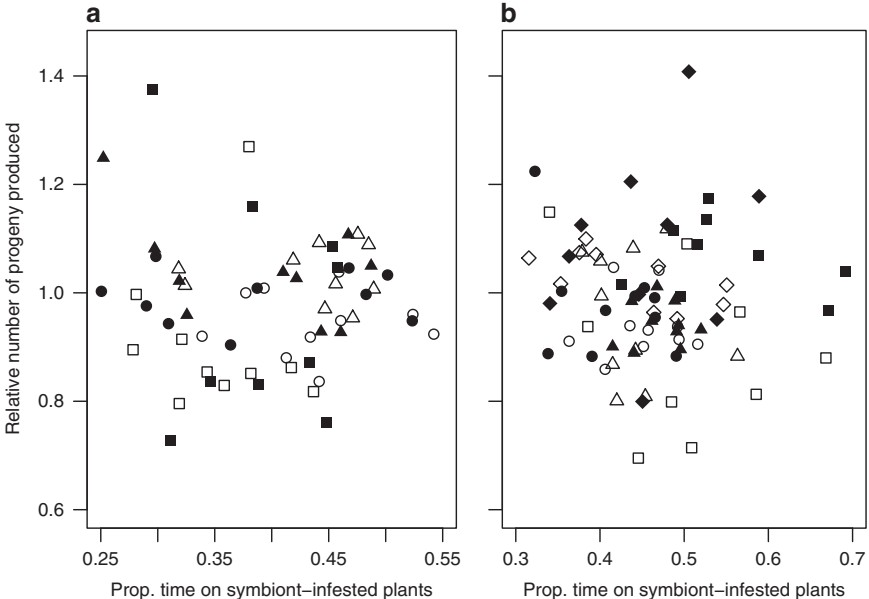

**Fig. 2** Relationship between aphid vigour and parasitic wasp response. Relationship between the relative number of progeny produced by symbiont-carrying and symbiont-free *Acyrthosiphon pisum* aphids on *Vicia faba* plants, and the mean proportion of time spent by parasitic wasps *Aphidius ervi* above symbiont-infested plants. **a** Tests with the symbiont *H. defensa*. These test include natural symbiont infections where the symbiont was removed with antibiotics in strains 101 (empty squares), 132 (solid squares) and 302 (empty triangles); artificial injections of strains 132 (solid triangles) and 404 (empty circles); test for systemic release of volatiles with an artificial injection of strain 132 (solid dots). **b** Tests with the other symbiont species. These include natural *Regiella insecticola* infections where the symbiont was removed with antibiotics in strains 319 (empty squares) and 126 (solid squares), and artificial injections of *R. insecticola* strains 319 (empty triangles) and 313 (solid triangles), *Spiroplasma* strains 227 (empty circles) and 237 (solid dots), *Serratia symbiotica* strain 619 (empty diamonds) and *Rickettsiella* strain 620 (solid diamonds). Spearman's rank correlation tests for these relationships were never significant as shown in Supplementary Table 1

secondary symbionts. No significant differences in plant attractiveness to parasitoids were seen with the natural association but in the experiments with introduced *Regiella* there was a preference for plants that had been fed upon by aphids without the symbiont (Fig. 5, columns i–iv). We injected two *Spiroplasma* isolates into secondary symbiont-free aphids and found that wasps showed a significant preference for plants previously attacked by aphids without this symbiont (Fig. 5, columns v–vi). In an experiment with a single naturally occurring isolate of *Serratia symbiotica*, wasps showed a significant preference for plants that had been fed on by aphids without the symbiont (Fig. 5, column vii). Finally, in an experiment with a single-injected isolate of *Rickettsiella* sp. wasps were also attracted to plants fed on by aphids without secondary symbionts (Fig. 5, column viii). There was no correlation between aphid vigour (measured by relative progeny production) and wasp preference suggesting that the results are not affected by any influence of the symbiont on the damage caused by the aphid to the plant (Fig. 2b, Supplementary Table 1).

## Discussion

Our study shows that plants infested with aphids carrying the symbiont *H. defensa* were less attractive to the parasitic wasp *A. ervi* through changes in herbivore-induced plant volatiles. We demonstrate this in a two-chamber olfactometer with a set of different *H. defensa* strains and aphid clones, and for three different *A. pisum* host plants. Reduced wasp attraction was not due to any chemical residue from the aphids remaining on the leaf, and the symbiont interfered with parasitic wasp recruitment through aphid-induced plant volatiles that were emitted systemically. In a population cage experiment, we demonstrate that changes in wasp attraction translate into reduced parasitoid attacks and hence increased fitness when aphids carried the

symbiont. We therefore provide evidence of a previously unknown mechanism through which symbionts protect their aphid hosts from parasitic wasps.

Many strains of *H. defensa* provide protection from parasitic wasp attack[19, 20] but even if the host survives, its fitness is reduced relative to unparasitised individuals[30]. By undermining the ability of the host plant to recruit parasitic wasps the symbiont provides an added level of protection that may result in the avoidance of parasitic wasp attack. Our results can be explained by the symbiont disrupting the blend of herbivore-induced plant volatiles produced by the host plant so that it no longer signals the presence of a host to the parasitic wasp. Alternatively, the modified blend might signal the presence of a well-defended host, which is uneconomical for the wasp to attack in terms of potentially wasted eggs or wasted time. This latter explanation, however, is less likely because we found that non-protective symbiont species as well as non-protective *H. defensa* strains also caused the plant to be less attractive to wasps. We have also found that total volatile emissions were significantly reduced by the presence of *H. defensa*. A possible interpretation of this result is that aphid symbionts reduce parasitic wasp recruitment by suppressing signalling pathways downstream of the production of multiple volatile compounds. This may be particularly efficient in avoiding attacks by generalist natural enemies like polyphagous aphid predators, which relative to specialist enemies often use more general cues to locate their hosts[31]. It would be interesting to carry out experiments exposing treated plants to a complex community of natural enemies in the field to explore this question further.

Many plants respond to herbivory by the emission of specific mixtures of volatiles that attract natural enemies[10]. Plant defences are often triggered by specific elicitors in herbivore oral secretions[32]. Insects, however, have evolved strategies to overcome

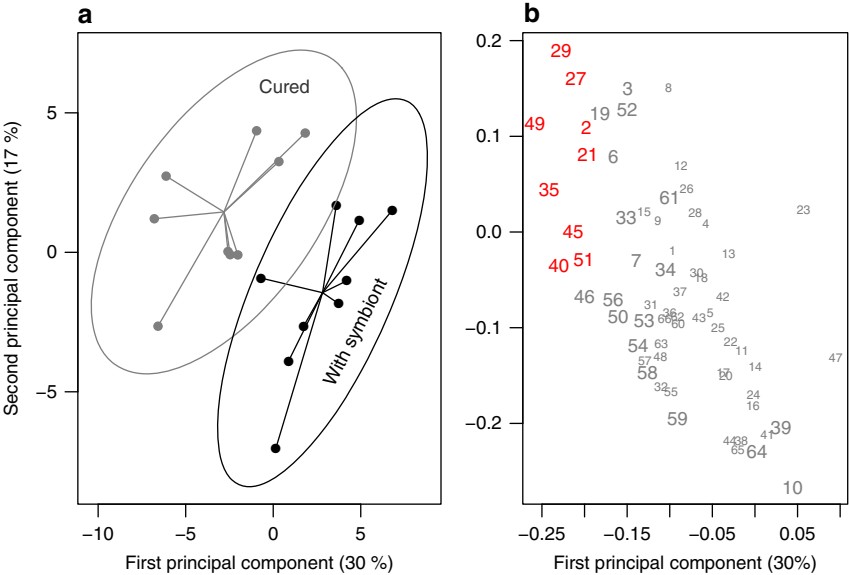

**Fig. 3** Discriminant analysis of the volatiles collected in the plant headspace. Volatiles were collected from *Vicia faba* plants infested with *Acyrthosiphon pisum* aphids (strain 101) with or without the symbiont *Hamiltonella defensa* (antibiotic curing). **a** First two principal components of a PLS-DA plot. Each data point represents a plant, the centre of the star is the multivariate centroid, and the circle the 95% confidence interval. **b** PLS-DA loading plot with all compounds depicted with respect to the first two principal components. Compounds depicted in a larger font have a variable importance in projection (VIP) score larger than 1, and those in red are emitted in a significantly larger amount by plants induced with symbiont-free aphids (Supplementary Table 2). In this figure some overlapping points were slightly displaced to increase clarity

these defences using salivary effectors[33–36] or by mimicking plant hormones[37]. Although plants infested with aphids carrying the symbiont or without it emitted the same volatile compounds, we found quantitative differences in the volatile blends. More specifically, the quantities of nine volatile compounds were significantly lower from plants in the symbiont treatment. Among these, *β*-cubebene and α-amorphen have previously been shown to be emitted by plants infested by aphids, but not by healthy plants[15], which makes these two compounds potential attractants of the specialist wasp *A. ervi*. Studies combining gas-chromatography, electro-antennography and behavioural assays may help to identify the characteristics of blend components that influence the behaviour of the parasitic wasp. Further research is also needed to understand the molecular mechanisms through which symbionts in the aphid are able to affect plant physiology, possibly through changes in the phytohormone levels.

Products derived from the obligate symbiont *Buchnera aphidicola* have been found in the saliva of *A. pisum*, and are known to induce plant defences[38]. Plant defensive responses mostly depend on the pathways regulated by phytohormones and studies with whiteflies and the Colorado potato beetle have demonstrated that symbionts can downregulate levels of the phytohormone jasmonic acid in ways that benefit their hosts[5, 6]. It would be interesting to investigate if compounds derived from aphid facultative symbionts are present in the saliva and injected into the plant. The complete genome of *H. defensa* is available and it contains sequences similar to those coding for effector proteins in plant pathogens that have been implicated in plant recognition of bacterial pathogens, which can potentially play a role in manipulating plant phytohormonal responses[39]. As reviewed by Pineda et al.[40], the microbial symbionts of plants may also modulate the production of herbivore-induced plant volatiles by altering plant defensive responses, which underlies the importance of microbial influences on plant-insect interactions.

Most of the secondary symbionts of pea aphids provide some conditional fitness benefits for their hosts, chiefly in combatting abiotic or biotic challenges. Early work suggested that different

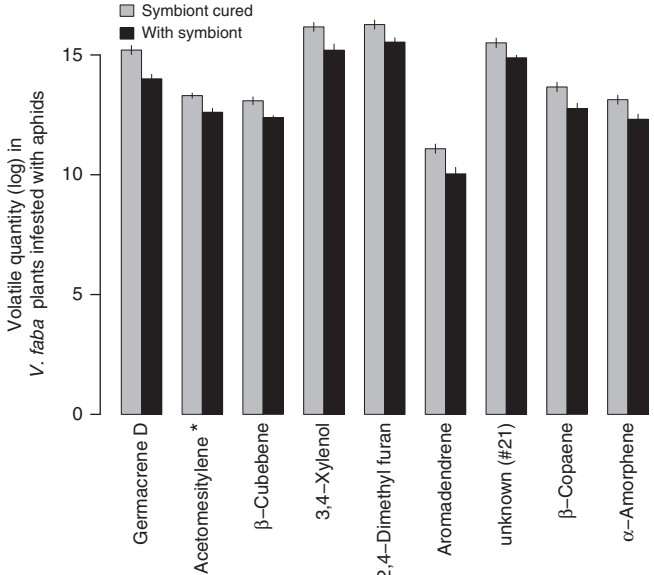

**Fig. 4** Quantification of selected volatile compounds found in the plant headspace. Volatiles were collected from *Vicia faba* plants infested with *Acyrthosiphon pisum* aphids carrying the symbiont *Hamiltonella defensa*, or not. Values are expressed as peak areas divided by dry plant weight (g) and the bars show standard errors. As shown in Supplementary Table 2, pairwise comparisons were performed for those compounds with a 'variable importance in projection' (VIP) score larger than one in the Principal least squares discriminant analysis (PLS-DA). Only compounds whose quantity was significantly different after correcting *p*-values with the false discovery rate approach are included. *Acetomesitylene: 1,3,5-Trimethyl-2-acetylbenzene

symbionts had specific functions, *H. defensa* and *R. insecticola* in defence against parasitic wasps and fungal pathogens, respectively,[20, 25] and *S. symbiotica* against heat shocks[28]. More recent research has revealed a more complicated picture with some

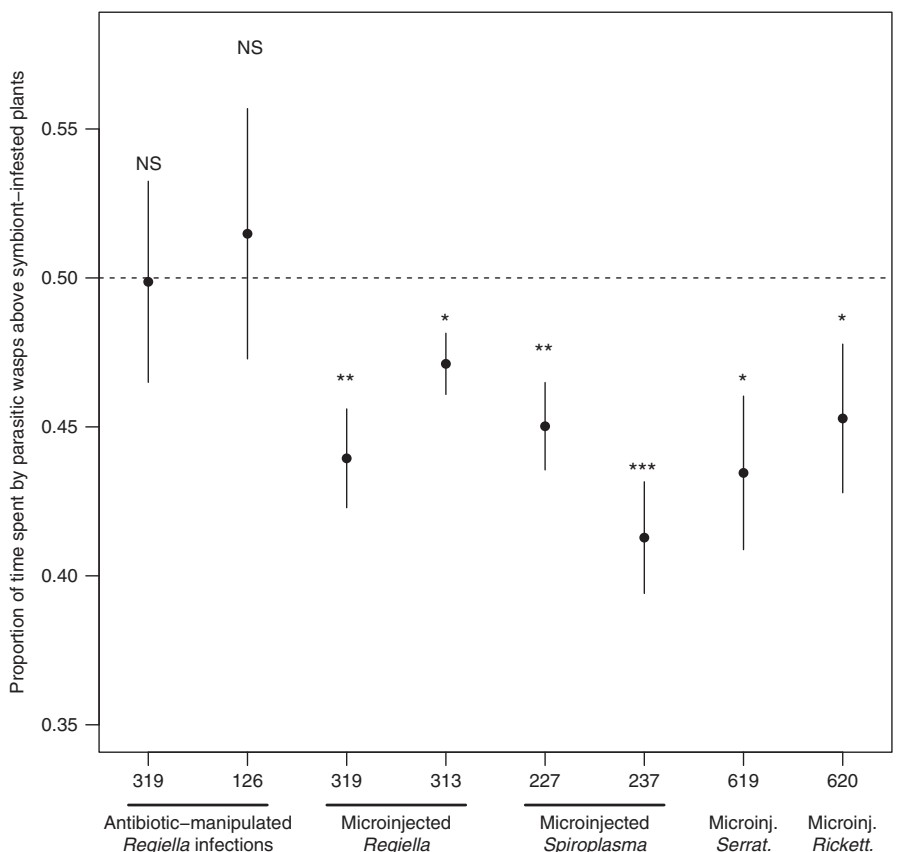

**Fig. 5** Parasitic wasp preference for volatiles from plants infested with aphids with or without different endosymbiont species. Parasitic wasp (*Aphidius ervi*) response for volatiles emitted by *Vicia faba* plants infested with aphids (*Acyrthosiphon pisum*) with or without different symbiont species. For each symbiont strain tested, the bars show standard errors and the asterisks the significance of the deviation from no choice (*t*-test: NS: non-significant; *$p < 0.05$; **$p < 0.01$; ***$p < 0.001$). *Regiella insecticola* ($n = 10$; column i—cured 319: $t_9 = -0.04$, $p = 0.4851$; ii—cured 126: $t_9 = 0.35$, $p = 0.6340$; iii—microinjected 319: $t_9 = -3.65$, $p = 0.0026$; iv—microinjected 313: $t_9 = -2.81$, $p = 0.0102$), *Spiroplasma* ($n = 10$; column v—microinjected 227: $t_9 = -3.40$, $p = 0.0039$; vi—microinjected 237: $t_9 = -4.65$, $p = 0.0006$), *Serratia symbiotica* ($n = 10$; column vii—microinjected 619: $t_9 = -2.54$, $p = 0.0158$) and *Rickettsiella* ($n = 10$; column viii—microinjected 620: $t_9 = -1.89$, $p = 0.0456$). Natural symbiont infections were removed with antibiotics in the *Regiella* strains 319 and 126, while the others were artificially microinjected into an aphid clone naturally lacking any secondary symbiont, and which was collected on *Lathyrus pratensis*

**Table 1 Symbiont strains used, and the aphid *Acyrthosiphon pisum* biotype from which they were eliminated with antibiotics, or obtained when microinjected**

| Symbiont species | Symbiont strain | Aphid biotype | Experiment |
|---|---|---|---|
| *Hamiltonella defensa* | 101 | *Ononis spinosa* | 1, 2, 3 |
| *H. defensa* | 132 | *Lotus pedunculatus* | 1 |
| *H. defensa* | 302 | *Medicago sativa* | 1 |
| *H. defensa* | 404 | *L. pedunculatus* | 1 |
| *Regiella insecticola* | 319 | *Trifolium pratense* | 4 |
| *R. insecticola* | 126 | *T. pratense* | 4 |
| *R. insecticola* | 313 | *T. pratense* | 4 |
| *Spiroplasma* | 227 | *M. sativa* | 4 |
| *Spiroplasma* | 237 | *M. sativa* | 4 |
| *Serratia symbiotica* | 619 | *Lathyrus odoratus* | 4 |
| *Rickettsiella* | 620 | *Pisum sativum* | 4 |
| None (recipient) | NA | *Lathyrus pratensis* | 1, 4 |

Information is given on the use of the different strains in each experiment. Microinjected symbionts were always injected into the same *A. pisum* lineage, which was obtained from *Lathyrus pratensis* plants

strains of a species failing to confer the expected advantage and in the case of *H. defensa* different isolates protecting against different wasp species[19]. In addition, new symbiont associations have been discovered with phenotypic effects that overlap with those previously investigated: different strains of *Spiroplasma* sp. associated with the pea aphid have particularly variable effects on its host[27]. It is thus not too surprising that the four additional symbionts we tested also influenced parasitic wasp recruitment. In the case of *R. insecticola*, the variation in response found is also not surprising because symbiont effects may depend on the symbiont strain, as well as on its interaction with the genotype of the aphid[30]. Further work on the mechanisms through which symbiont presence can affect the emission of plant volatiles may reveal the degree to which this phenomenon is a specifically selected adaptation or a byproduct of other processes through which the symbiont influences plant physiology, and whether the different symbionts influence volatile emission through a single or multiple mechanisms.

In conclusion, we show that microbial symbionts add another level of complexity to the already intricate role of plant volatiles in mediating the relationship between plants, herbivores and their natural enemies. We also demonstrate that they protect the host (and themselves) from parasitoid wasp attack both directly, by reducing the chances of successful wasp development, and indirectly, by reducing the probability of wasp attack. Aphids

include some of the most important pests of temperate crops, and understanding these relationships may assist in the challenge of designing more environmentally benign pest control strategies. These strategies may include assessing the prevalence of protective symbionts in pest populations[41], and selection of plant varieties that once attacked by aphids, maximise the attraction to aphid natural enemies[42].

## Methods

**Experimental organisms.** All aphids used in this study (Table 1) were collected from different host plant species in Oxfordshire (southern England) and maintained in the laboratory on broad bean plants (*V. faba*, cv. the Sutton) at constant conditions of $20 \pm 1 \,°C$ and $70 \pm 5\%$ relative humidity with a 16:8 h light:dark (L:D) regime, to assure continuous asexual reproduction. After collection, aphids were screened using diagnostic PCRs for the eight common facultative endosymbionts known from this aphid species[43–45]. Symbionts were removed from naturally infected clones using a cocktail of antibiotics (ampicillin, cefotaxime and gentomicin) administered through the host plant. These antibiotics do not harm the obligate primary symbiont *Buchnera aphidicola*. New symbiont infections were created by microinjecting haemolymph from a donor aphid into a receiver clone which carried no natural facultative symbionts. All experiments were performed at least 10 aphid generations after manipulation. Before the experiments symbiont composition was checked using diagnostic PCRs. Further details of the experimental procedures can be found in refs. [18, 46,], while a summary of the aphids and symbionts used in this study is provided in Table 1.

*Aphidius ervi* parasitic wasps were obtained from Koppert Biological Systems (Berkel en Rodenrijs, The Netherlands) and maintained on an *A. pisum* clone that is naturally free from any facultative symbiont and which was not used in any of the behavioural experiments. Wasps used in the experiments were 2- to 3-day-old mated females, which for the 24 h prior to the experiment had been provided with diluted honey (10% solution) and aphids so that they could gain oviposition experience as in ref. [47]. The hosts were withdrawn two h before the experiment.

**Effect of symbiont *H. defensa* on parasitic wasp attraction.** The effect of *A. pisum* symbionts on the production of herbivore-induced plant volatiles and wasp preference was tested in a two-chamber olfactometer[48]. This olfactometer consists of a Perspex cylinder, which is divided longitudinally into two identical compartments. The cylinder is placed in a vertical position and the top end is closed off with thin mesh. A test plant is placed in each compartment, and parasitic wasps are released in an enclosed space above the mesh. Wasp preference for either of the two plants is assessed as the proportion of time wasps spent on top of each chamber. To trigger the induction of herbivore-induced volatiles by *V. faba* plants, a single plant was planted in a 1.1 L pot. Then, at the 2–3 leaf developmental stage (2-week-old plants), two clip cages containing 10 wingless adult female aphids each were placed on the two halves of the bottommost leaf. Aphids were allowed to feed on the plant for 5 days, and then removed. Every other day, aphid offspring was removed from the clip cage and counted as a measure of aphid vigour and feeding intensity. The protocol was slightly different for experiments with the plants *Ononis spinosa* and *Lotus pedunculatus*. Instead of using clip cages, 20 wingless adult females were placed onto a 6-week-old plant planted in a 1.1 litre pot. Plants were individually covered with a micro-perforated plastic bag to prevent aphids from escaping and the insects allowed to feed for 5 days before they were removed from the plant.

All tests were performed by comparing the attractiveness of plants which had been fed upon by genetically identical, clonal aphids, which differed only in their symbiont status. To prevent volatiles from the soil interfering with the behaviour of the parasitic wasp, pots were covered with aluminium foil. Five minutes after placing the plants in the olfactometer, a single parasitic wasp was released in the centre of the arena and after a minute's pause its behaviour was recorded for the following 6 min. Wasps that did not forage during the 4 min following their release were discarded. In each bioassay testing a specific symbiont strain or plant, the response of five female wasps was monitored. The mean response of these five wasps was considered a replicate, and this was repeated 10 times, each time with a new set of plants and wasps. The relative position of plants with symbiont-carrying and symbiont-free aphids was changed after every third replicate when the olfactometer was left open to allow any volatiles to disperse. The person recording the behaviour of the wasp was unaware of the treatment allocation to the two chambers. The effectiveness of the two-chamber olfactometer used here was assessed in preliminary tests with a limited number of plant replicates ($n = 4$). These tests showed that relative to plants that carried an empty clip cage, plants carrying a clip cage with aphids were more attractive to wasps.

We first compared the attraction of plants that had been fed on by aphids that naturally carried *H. defensa* with the same clone from which the symbiont had been removed (strains 101, 132 and 302, Table 1). We then repeated the experiment twice comparing an aphid clone that when collected from the field carried no facultative symbionts with the same clone into which one of two *H. defensa* isolates had been introduced by microinjection (strains 132 and 404, Table 1). With the exception of strain 101, these symbiont strains are known to confer direct protection against *A. ervi* in the laboratory[19]. To investigate if volatiles

are produced systemically, one of the experiments with an introduced *H. defensa* isolate (strain 132) was repeated but with the leaf on which the aphids had fed covered with aluminium foil (Supplementary Fig. 1). Carrying symbionts may be costly and so reduce aphid vigour, and hence possibly cause less feeding damage to the plant and lower volatile emission. To control for this we repeated an experiment with the strains 132 and 101 (Table 1) but with double (20) the number of symbiont-bearing compared to symbiont-free aphids. Finally, we repeated two of the experiments with natural *H. defensa* infections but with the aphid feeding on the host plant species from which they were collected [*Ononis spinosa* (strain 101) and *Lotus pedunculatus* (strain 132)]. The pea aphid taxon is composed of host-adapted races or biotypes and microsatellite analyses had shown that these clones belonged to the biotypes associated with the two host plants[24].

**Effect of symbiont *H. defensa* on parasitic wasp attack.** To test whether differences in parasitoid recruitment translate into differences in parasitism, we compared the attack rates experienced by secondary symbiont-free aphids (sentinel aphids) feeding on plants that had previously been fed on by aphids with or without *H. defensa* (strain 101, 15 replicates per aphid type). The experiments were carried out in cubic gauze cages of $47.5 \times 47.5 \times 47.5$ cm (BugDorm 44545 F, Taichung, Taiwan), which are arenas large enough for parasitoid wasps to show typical searching and oviposition behaviours[47]. Plants were prepared by allowing aphids of either of the two types to feed on them for 5 days before they were removed. Then, 30 genetically identical "sentinel" aphids were placed on each plant. When attacked by *A. ervi*, pea aphids drop from the plant and disperse within the cage. To identify which treatment they came from, each plant in the same cage received a different coloured aphid clone: one red and the other green. To control for any wasp preference, clone colour was stratified across treatments. The aphids were 3 days old when used and highly susceptible to parasitoid attack. A plant of each type was placed at opposite corners of the cage and 1 h later two mated female *A. ervi* wasps with oviposition experience were released in the centre of the cage. Wasps were allowed to search for and attack aphids for 2 h before being removed. The aphids were then collected and reared at 18 °C in Petri dishes on healthy leaves of *V. faba* with their petioles inserted into 2% agar gel to keep them fresh. Ten days later the number of parasitic wasp mummies and adult aphids obtained from each plant was counted.

**Effect of symbiont *H. defensa* on volatile composition.** We explored the effect of the symbiont *H. defensa* (strain 101) on the composition of volatiles in the headspace of *V. faba* plants previously infested with *A. pisum* aphids. Nine replicate plants that had been exposed to aphids with or without *H. defensa* were prepared as described above. Once aphids were removed, dynamic headspace sampling of volatiles was carried out in a climate chamber at $20 \pm 1 \,°C$. The soil and plant roots were carefully wrapped with aluminium foil to exclude any volatiles not coming from the plant. The plants were then individually placed into 2.5 l glass jars connected to an air flow. Humidified air, mixed with $CO_2$ at 400 p.p.m., was supplied to each jar. After 30 min, volatile collection started by drawing air out of the glass jar at a rate of 300 mL min$^{-1}$ through a stainless steel tube filled with 200 mg Tenax TA filter (20/35 mesh; CAMSCO, Houston, TX, USA) for 4 h. Volatiles from a total of six glass jars were collected at the same time. Volatiles from test plants were collected in blocks of five and the sixth jar was kept as a control containing only a pot with soil wrapped in aluminium foil.

The volatiles were analysed as described in ref. [49]. A Thermo Trace Ultra Gas Chromatograph, in combination with a Thermo Trace DSQ quadrupole Mass Spectrometer (Thermo Fisher Scientific, Waltham, USA), were used for separation and detection of plant volatiles. Before analysis, moisture was removed from the Tenax adsorbent material by flushing with nitrogen (50 mL min$^{-1}$) for 10 min. The volatiles were then released from the Tenax filter using a helium flow of 20 mL min$^{-1}$ for 10 min under an Ultra 50:50 thermal desorption unit (Markes, Llantrisant, UK) at 250 °C. At the same time, volatiles were re-collected in a universal solvent trap Unity (Markes) at 0 °C. The volatiles were then released and transferred to a ZB-5MSi analytical column [30 m × 0.25 mm I.D. × 0.25 μm F.T. with a 5 m built-in guard column (Phenomenex, Torrance, CA, USA)] by heating the solvent trap for an incremental 40 °C every second until a temperature of 280 °C was reached and then held for 10 min. The gas chromatograph oven was initially set at a temperature of 40 °C for 2 min before being raised by 6 °C a minute until it reached 280 °C at which it was kept for 4 min. The DSQ mass spectrometer operated in a scan mode with a mass range of 35–400 amu at 4.70 scans s$^{-1}$ and the spectra were recorded in electron impact ionisation (EI) at 70 eV. The mass spectrometer transfer line and ion source were set at 275 and 250 °C, respectively.

Compounds were identified by comparing mass spectra data with those in the NIST 2005 and the Wageningen Mass Spectral Database of Natural Products MS libraries. Some compounds were also identified through linear retention indices based on the time they eluted from the gas chromatograph column relative to standard compounds. A target (single) ion for each compound was used for the measurement of peak area. Volatile samples from control jars (i.e., with just the pot and the soil wrapped in aluminium foil) were considered as blank samples. Volatiles recorded in these samples were thus treated as non-plant-related artefacts and subsequently excluded from the dataset.

**Effect of other symbionts on parasitic wasp recruitment**. The response of wasps to plants previously attacked by aphids carrying these different symbionts was assessed as in Experiment 1 with details of the aphid-symbiont strains involved given in Table 1. As in previous experiments, for *Regiella* strains 319 and 126 natural symbiont infections were removed with antibiotics, whereas the others were artificially microinjected into an aphid clone naturally lacking any secondary symbiont, and which was collected on *Lathyrus pratensis*.

**Statistical analyses**. All analyses were performed in R 3.3.1 (R development Core Team). Wasp preference in the two-chamber olfactometer followed a normal distribution and was analysed using *t*-tests with the null hypothesis being equal time allocation to the two treatments. The mean time allocation of the five wasps tested on a single plant pair served as the response variable in the analyses. Each test was repeated 10 times with new plants and new wasps. Spearman's rank correlation was used to test the relationship between the relative number of progeny produced by symbiont-bearing and symbiont-free aphids and wasp preference in the olfactometer. Differences in parasitism in the population cage experiment were explored with a paired *t*-test.

Plant volatile quantity (peak areas corrected by dry plant weight in grams) was log transformed before PLS-DA using the function plsda from the mixOmics package[50]. The significance of the treatment was assessed using a permutation analysis (9999 repetitions) implemented in the MVA.test from the RVAideMemoire package[51]. Variable importance in projection (VIP) scores calculated using PLSDA.VIP from the RVAideMemoire package were used to identify compounds important in treatment separation[52], which were then compared using *t*-tests after log transformation. In these comparisons *p*-values were corrected for multiple comparisons based on false discovery rates as implemented by the R function p.adjust. Relative to family-wise methods like Bonferroni, the false discovery rate method is less stringent in controlling type I errors, and is therefore more appropriate when a large number of comparisons is performed and some false positives are acceptable[53]. Since a different ion was used to quantify the various volatile compounds, total emissions cannot be obtained by summing up the amounts of the compounds obtained. Therefore, to test whether the symbiont had an overall effect on volatile emissions, a non-parametric two-sample sign test was used using the function SIGN.test from the BSDA package[54].

**Data availability**. The data sets generated during and/or analysed during the current study are available from the corresponding author on reasonable request.

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

## Acknowledgements

We are grateful to Koppert Biological Systems for kindly providing parasitic wasps. E.F. was funded by Marie Curie Intra-European Fellowship within the Seventh European Community Framework Programme, FP7-PEOPLE-2012-IEF #329648, the Regional Council of Reunion, the Departmental Council of the Region Reunion, the European Union (EAFRD) and CIRAD. A.M. and H.C.J.G. acknowledge funding from the UK's Natural Environment Research Council (NE/K004972/1). M.D. and E.F. are grateful to the COST Action FA1405 for fruitful discussions and inspiration within this collaborative network.

## Author contributions

R.G. and E.F. conceived and designed the research. E.F. wrote the Marie Curie project with input from H.C.J.G., R.G. and M.D. M.M. and C.Y. performed the behavioural experiments supervised by A.M., R.G. and E.F. B.T.W. collected, quantified and identified plant volatiles. A.M. created and characterised pea aphid lines. E.F. analysed data. H.C.J.G., R.G., M.D. and E.F. wrote the manuscript, and all authors contributed with revisions.

## Additional information

**Competing interests:** The authors declare no competing financial interests.

