## [Peer Review File · Nature Communications]

Reviewers' comments:

Reviewer #1 (Remarks to the Author):

This paper presents a series of experiments addressing the hypothesis that endosymbionts in aphids cause plant volatile emission to change, which causes parasitoids to be less attracted to the plant, leading to reduced parasitism of the aphids. The results appear to support the hypothesis, and as far as I know this endosymbiont-plant-parasitoid connection through volatile production has not been shown before. The experiments are well designed and it is that they include behavior of the wasp, chemistry of the plant, and resulting parasitism of the aphids, and a range of different symbionts.

The methods, results most of the discussion are clearly written. The substantive comments I have are mostly about the abstract and introduction. The abstract was a bit misleading and the introduction needed some structure and motivation with respect to the set of experiments that were conducted.

Line 4-5 "... while insect herbivores often carry..." This simplification is misleading. They often carry endosymbionts that improve fitness in various ways, but there has not been enough research to say that they often carry endosymbionts that improve survival after attack by a natural enemy in general. There have been a few examples, mostly in aphids, but most of the examples are of microbial disease, and endosymbionts have also been shown to decrease resistance to enemies.

Line 8-10. The sentence starting "We show that..." It seems like having this relationship between the endosymbiont is explanation enough without the statement above saying that the endosymbiont improves survival of the host after attack.

Lines 48-50 The sentence starting "Here the symbiont reduces."

The purpose of this sentence isn't clear

- what about parasitic symbionts?

-aren't facultative symbionts generally not always associated with the host.?

Line 55 be explicit about what "This idea" is. I can see how the series of experiments below are related to the topic of the sentence above, but it would be good to have an explanation about the motivation and expectation or hypothesis being tested for each one. As it is now, they seem interesting, but kind of a disorganized basket of related experiments.

Line 63-64- later in the ms I understood what you wanted to control for, but here it was confusing. I wondered what you were controlling.

Line 68 -What is a population cage?

Line 73- the same wasp species

Lines 92-93- How did you now for sure that they had been cured?

Line 105- In the experiments generally the replication isn't clear, or isn't stated at all. They should be in the methods, and also it would be nice to have the replication near the lines on figure 1.

Line 116- were instead of was

Line 147-140- a reference for the microsat analysis

Line 274- explain "difference in the opposite direction"

Line 310- "no longer signal a presence" This makes me curious about how well the volatile mix matches an un-infested plant.

Lines 311-312- Are there aphid hosts that are so well defended that *A. ervi* would be unsuccessful?

Lines 314-316- This is a nice idea, but needs to be explained some more. For example, many readers may not know what Batesian mimicry is.

Lines 353-354- change is sentence because we already know that they add another layer of complexity to the relationship. This paper adds another way in which they add another layer of complexity.

Lines 354-355- It isn't clear what "resolving this relationship" means

Lines 355-356- elaborate.

Reviewer #2 (Remarks to the Author):

A very interesting manuscript on how symbionts affect the recruitment of parasitic wasps by plants, adding to a growing knowledge of complex multi-trophic interactions. The findings are of interest both to fundamental as well as applied biology, the authors show how the symbiont via the aphid affects the plant's ability to recruit parasitic wasps for defence, opening up avenues to study the mechanisms by which this interaction takes place. In addition, this is of interest to applied agricultural sciences where more sustainable crop protection solutions frequently focus on the recruitment on beneficial insects into the agro-ecosystem. The manuscript is well written.

The following questions and comments arose on reading the manuscript:

Methodology:

Experiment 1

Line 120: How were the control plants treated? Could volatile changes be due to stress associated with clip cages or different environment inside perforated plastic bags? Indeed were there differences between these two control treatments? I gather from the methodology that there was not a positive control, i.e. an aphid free plant + an aphid free plant with a clip cage/plastic bag + aphid infested w/symbiont, aphid infested w/out symbiont. How does this affect the results?

Line 130: I am unclear whether every replicate had a fresh plant or were they pseudo-replicates and was this accounted for in the analysis?

The changes in volatile profiles between the treatments seem to be quantitative rather than due to an absence or presence of different volatile compounds. Is this correct? I think this warrants a discussion in the manuscript.

No electrical antennography studies accompany the data. This, and accompanying olfactory bioassays on the identified chemicals would enhance the manuscript by identifying the specific compounds which elicit the behavioural responses by the parasitoids, leading to underlying mechanisms by which the symbiont affects the plants ability to recruit defence. Was this not an option for the authors?

In addition the following minor comments need addressing:

Line 41: expand briefly on how symbionts manipulate induced defences in the host plant.

Line 84: refer to table 1 here for the host plant species.

Line 194: add 'for' to 'was kept 4 min'.

Discussion would benefit from more detail in places, it's not enough for this audience to know that things can change, they already know that, but what changes is important to them.

Line 330: change phytohormone levels 'in the plant' in ways that... plus expand on which hormones.

Line 333: add more detail, which proteins have been implicated?

- How does this relate to the findings of this work? Are there any notable commonalities or differences.

Line 335: The 'how' would be interesting here, did Pineda et al 2010 report on which symbionts and what they changed in the plant?

The conclusion does not do the work justice.

Reviewer #3 (Remarks to the Author):

This research article, which focuses on how symbionts may alter parasitism of their hosts by altering host plant parasite-attracting chemicals, is comprehensive and generally well-written. Particularly nice are how alternative hypotheses are laid out in the beginning. Based on limited work in other systems on plant-symbiont-host interactions, one could predict that symbiont alteration of plant chemicals, if it occurs, could either compliment protection conferred via enhanced host resistance to parasitoids or could counteract such resistance-based protection. Either finding would be novel and interesting. In this case, the researchers find that plants that have been fed on by hosts (aphids) with symbionts release fewer chemicals, and that the parasitoids are less attracted to these plants. The lesser attraction to plants that have been fed on by hosts with symbionts is consistent across host-symbiont pairings used in several experiments. This is a novel finding, and of interest from a basic sciences and applied perspective.

One challenge of this manuscript centers around Table 1. These are a complex set of experiments that use different aphid-symbiont combinations. The methods repeatedly refer to Table 1 as a reference of what lines were used, but from Table 1, you cannot determine which experiments used which lines. I finally gave up and decided I would tease it apart during the results. In fact, you can't determine it from the text of the results either and have to turn to the figures to determine what lines the experiments used. I like the simplicity of the text as written, so one solution is to add more details in the table about which lines were used for which experiments.

No statistics are presented in the first paragraph or last paragraphs of the results, but they are presented in other paragraphs.

In Table S1, what does systemic response mean?

Were assumption of t tests met? Were they checked?

Reply to Associate Editor:

Dear Dr Frago,

Your manuscript entitled "Symbionts protect aphids from parasitic wasps by attenuating herbivore-induced plant volatiles" has now been seen by three referees. You will see from their comments below that while they find your work of interest, some important points are raised. We are interested in the possibility of publishing your study in *Nature Communications*, but would like to consider your response to these concerns in the form of a revised manuscript before we make a final decision on publication. We therefore invite you to revise and resubmit your manuscript, taking into account the points raised. We particularly encourage you to address Reviewer 2's comment about reinforcing the findings with electrical antennography studies or olfactory bioassays. Please highlight all changes in the manuscript text file. We are committed to providing a fair and constructive peer-review process. Do not hesitate to contact us if you wish to discuss the revision in more detail or if there are specific requests from the reviewers that you believe are technically impossible or unlikely to yield a meaningful outcome.

The objective of our study was to show that aphid symbionts manipulate herbivore-induced plant volatiles for the benefit of their host, the aphid. We provide evidence in terms of plant chemistry and insect behaviour to support our conclusions. We believe the proposed additional experiments would not change these conclusions. The proposed additional electrical antennography experiments, for example, would provide evidence that the chemoreceptors on the parasitoids' antennae perceive the chemicals, but this is something that is already known. One of the reasons we decided to work with the model system *Vicia faba* (plant), pea aphid and *Aphidius ervi* (parasitic wasp) is that the plant volatiles responsible for parasitoid attraction have already been studied in detail elsewhere using olfactometry, wind tunnel and antennogram tests (see references below). In addition to the identification of aphid-induced plant volatiles, these studies show that parasitoid responses to individual blend components often cannot be extrapolated to responses to complex blends. This would make antennography experiments technically very challenging. Revealing the underlying mechanism explaining volatile-mediated foraging behaviour at the level of sensory neurophysiology is a study topic in itself and is not further explored here. We acknowledge, however, that the information presented in the works exploring parasitic wasp responses to plant volatiles was not clearly included in our manuscript, something that we have now included in this revised version.

Du Y-J, Poppy GM, Powell W (1996) Relative importance of semiochemicals from first and second trophic levels in host foraging behavior of *Aphidius ervi*. *J Chem Ecol* 22:1591-1605. doi: 10.1007/BF02272400.

Du YJ, Poppy GM, Powell W, et al (1998) Identification of semiochemicals released during aphid feeding that attract parasitoid *Aphidius ervi*. *J Chem Ecol* 24:1355-1368. doi: 10.1023/A:1021278816970

Guerrieri E, Poppy GM, Powell W, et al (1999) Induction and systemic release of herbivore-induced plant volatiles mediating in-flight orientation of *Aphidius ervi*. *J Chem Ecol* 25:1247-1261.

Takemoto H, Takabayashi J (2015) Parasitic wasps *Aphidius ervi* are more attracted to a blend of host-induced plant volatiles than to the independent compounds. *J Chem Ecol* 41:801-807. doi: 10.1007/s10886-015-0615-5

Data availability statements and data citations policy: All Nature Communications manuscripts must include a section titled "Data Availability" at the end of the Methods section or main text (if no Methods).

We have included this section in the revised manuscript mentioning that data is available from the authors. If the manuscript is accepted for publication, data will be deposited in a public repository.

Reply to Reviewer #1

This paper presents a series of experiments addressing the hypothesis that endosymbionts in aphids cause plant volatile emission to change, which causes parasitoids to be less attracted to the plant, leading to reduced parasitism of the aphids. The results appear to support the hypothesis, and as far as I know this endosymbiont-plant-parasitoid connection through volatile production has not been shown before. The experiments are well designed and it is that they include behavior of the wasp, chemistry of the plant, and resulting parasitism of the aphids, and a range of different symbionts.

The methods, results most of the discussion are clearly written. The substantive comments I have are mostly about the abstract and introduction. The abstract was a bit misleading and the introduction needed some structure and motivation with respect to the set of experiments that were conducted.

We have made an effort to improve the introduction section and the abstract.

Line 4-5 "... while insect herbivores often carry..." This simplification is misleading. They often carry endosymbionts that improve fitness in various ways, but there has not been a enough research to say that they often carry endosymbionts that improve survival after attack by a natural enemy in general. There have been a few examples, mostly in aphids, but most of the examples are of microbial disease, and endosymbionts have also been shown to decrease resistance to enemies.

We agree with the reviewer, and we have rephrased the sentence "... while insect herbivores often carry..." by "... while insect herbivores may carry...".

Line 8-10. The sentence starting "We show that..." It seems like having this relationship between the endosymbiont is explanation enough without the statement above saying that the endosymbiont improves survival of the host after attack.

We have clarified that the symbiont improves aphid survival by preventing the development of the parasitic larva.

Lines 48-50 The sentence starting "Here the symbiont reduces." .The purpose of this sentence isn't clear, what about parasitic symbionts? Aren't facultative symbionts generally not always associated with the host?

We have rephrased this sentence.

Line 55 be explicit about what "This idea" is. I can see how the series of experiments below are related to the topic of the sentence above, but it would be good to have an explanation

about the motivation and expectation or hypothesis being tested for each one. As it is now, they seem interesting, but kind of a disorganized basket of related experiments.

We agree with the reviewer and we have added several new phrases to make our hypotheses more clear and specific.

Line 63-64- later in the ms I understood what you wanted to control for, but here it was confusing. I wondered what you were controlling.

We have added a sentence to clarify this.

Line 68 –What is a population cage?

We have added “Population cages are arenas large enough for parasitoid wasps to show typical searching and oviposition behaviours.”.

Line 73- the same wasp species

Added

Lines 92-93- How did you now for sure that they had been cured?

Symbiont composition was always checked before the experiments using diagnostic PCRs. We have added this information in the text.

Line 105- In the experiments generally the replication isn't clear, or isn't stated at all. They should be in the methods, and also it would be nice to have the replication near the lines on figure 1.

In each bioassay testing a specific symbiont strain or plant, the response of five female wasps was monitored. The mean response of these five wasps was then considered as a replicate, and this was repeated 10 times, each time with a new set of plants. We have clarified this information in the text.

Line 116- were instead of was

Corrected

Line 147-140- a reference for the microsat analysis

Added.

Line 274- explain “difference in the opposite direction”

We have rephrased this sentence.

Line 310- “no longer signal a presence” This makes me curious about how well the volatile mix matches an un-infested plant.

Volatile blends emitted by non-infested plants are usually quantitatively very different from

those of infested plants. As revealed by the papers exploring these volatiles in *Vicia faba* (see references above) many new volatiles are biosynthesized upon insect attack. The volatile blends reported in our work are therefore likely to be much more complex than those emitted by a healthy plant. We have modified this sentence to accommodate this reviewer's comment.

Lines 311-312- Are there aphid hosts that are so well defended that A. ervi would be unsuccessful?

Yes, there are. There are symbionts that confer 100% protection against the wasp as shown in the cited reference (McLean and Godfray, 2015).

Lines 314-316- This is a nice idea, but needs to be explained some more. For example, many readers may not know what Batesian mimicry is.

We have added extra information to this sentence, and a new reference.

Lines 353-354- change is sentence because we already know that they add another layer of complexity to the relationship. This paper adds another way in which they add another layer of complexity.

We agree with the reviewer and have added to this sentence that our study adds "another layer of complexity to the already intricate role of plant volatiles" in mediating these interactions.

Lines 354-355- It isn't clear what "resolving this relationship" means

We have clarified this sentence.

Lines 355-356- elaborate.

We have added a sentence suggesting potential strategies to improve natural enemy control over aphid pests.

Reply to Reviewer #2

A very interesting manuscript on how symbionts affect the recruitment of parasitic wasps by plants, adding to a growing knowledge of complex multi-trophic interactions. The findings are of interest both to fundamental as well as applied biology, the authors show how the symbiont via the aphid affects the plant's ability to recruit parasitic wasps for defence, opening up avenues to study the mechanisms by which this interaction takes place. In addition, this is of interest to applied agricultural sciences where more sustainable crop protection solutions frequently focus on the recruitment on beneficial insects into the agro-ecosystem. The manuscript is well written.

The following questions and comments arose on reading the manuscript:

Methodology:

Experiment 1

Line 120: How were the control plants treated? Could volatile changes be due to stress associated with clip cages or different environment inside perforated plastic bags? Indeed were there differences between these two control treatments? I gather from the methodology that there was not a positive control, i.e. an aphid free plant + an aphid free plant with a clip

cage/plastic bag + aphid infested w/symbiont, aphid infested w/out symbiont. How does this affect the results?

In our experiments control treatment refers to whether aphids were cured from endosymbionts or not. Thus on both plants offered in the bioassay, aphids had been contained in clip cages. We cannot exclude that there is an effect of the clip cages/plastic bag but this would be present in both the treatment and the control. However, preliminary tests with a limited number of replicates revealed that relative to plants that carried an empty clip cage, plants carrying a clip cage with aphids were more attractive to wasps. We have added this information in the text.

Line 130: I am unclear whether every replicate had a fresh plant or were they pseudo-replicates and was this accounted for in the analysis?

In each bioassay testing a specific symbiont strain or plant, the response of five female wasps was monitored. The mean response of these five wasps was then considered as a replicate, and this was repeated 10 times, each time with a new set of plants. We have clarified this information in the text.

The changes in volatile profiles between the treatments seem to be quantitative rather than due to an absence or presence of different volatile compounds. Is this correct? I think this warrants a discussion in the manuscript.

This is correct, we have added a sentence in the discussion that explains this.

No electrical antennography studies accompany the data. This, and accompanying olfactory bioassays on the identified chemicals would enhance the manuscript by identifying the specific compounds which elicit the behavioural responses by the parasitoids, leading to underlying mechanisms by which the symbiont affects the plants ability to recruit defence. Was this not an option for the authors?

Please see our reply to the Editor.

In addition the following minor comments need addressing:

Line 41: expand briefly on how symbionts manipulate induced defences in the host plant.

Added

Line 84: refer to table 1 here for the host plant species.

Added

Line 194: add 'for' to 'was kept 4 min'.

Added

Discussion would benefit from more detail in places, it's not enough for this audience to know that things can change, they already know that, but what changes is important to them.

Several new sentences have been added to the discussion section.

Line 330: change phytohormone levels 'in the plant' in ways that... plus expand on which hormones.

We have added that we refer to the phytohormone jasmonic acid

*Line 333: add more detail, which proteins have been implicated?
- How does this relate to the findings of this work? Are there any notable commonalities or differences.*

We have added that these are effector proteins used by plants to recognise plant pathogens, and potentially used by them to manipulate plant phytohormonal responses.

Line 335: The 'how' would be interesting here, did Pineda et al 2010 report on which symbionts and what they changed in the plant?

We have clarified that this work is a review, and added that the main message coming from it is that plant symbionts often alter plant defensive responses.

The conclusion does not do the work justice.

We have extended the conclusion, and made it more specific.

Reply to Reviewer #3

This research article, which focuses on how symbionts may alter parasitism of their hosts by altering host plant parasite-attracting chemicals, is comprehensive and generally well-written. Particularly nice are how alternative hypotheses are laid out in the beginning. Based on limited work in other systems on plant-symbiont-host interactions, one could predict that symbiont alteration of plant chemicals, if it occurs, could either compliment protection conferred via enhanced host resistance to parasitoids or could counteract such resistance-based protection. Either finding would be novel and interesting. In this case, the researchers find that plants that have been fed on by hosts (aphids) with symbionts release fewer chemicals, and that the parasitoids are less attracted to these plants. The lesser attraction to plants that have been fed on by hosts with symbionts is consistent across host-symbiont pairings used in several experiments. This is a novel finding, and of interest from a basic sciences and applied perspective.

One challenge of this manuscript centers around Table 1. These are a complex set of experiments that use different aphid-symbiont combinations. The methods repeatedly refer to Table 1 as a reference of what lines were used, but from Table 1, you cannot determine which experiments used which lines. I finally gave up and decided I would tease it apart during the results. In fact, you can't determine it from the text of the results either and have to turn to the figures to determine what lines the experiments used. I like the simplicity of the text as written, so one solution is to add more details in the table about which lines were used for which experiments.

We have added a new column in Table 1 that specifies in which experiment each clone was used. We have also included extra information on the strains used in the methods section.

No statistics are presented in the first paragraph or last paragraphs of the results, but they

are presented in other paragraphs.

In these paragraphs several comparisons are performed and we prefer to keep statistical tests in the figure legend.

In Table S1, what does systemic response mean?

We have made this legend more clear.

Were assumption of t tests met? Were they checked?

Model assumptions were checked, we have added this information in the text.

Reviewers' comments:

Reviewer #1 (Remarks to the Author):

The authors have made a sincere revision, and addressed all of the concerns that I had in a reasonable way. I agree with the authors that adding mechanistic antennography experiments would not add particularly useful information to the study.

Note that the changes made in the ms. were not highlighted in the text or listed by line number in the response to the reviewers so it took effort to evaluate if reasonable changes were made. One very minor question I still have- I had asked what a "population cage" is. The authors added to the text that "population cages are arenas large enough for parasitoids to show typical searching and oviposition behavior." What size is that?

Reviewer #2 (Remarks to the Author):

I disagree with the authors in terms of the usefulness and feasibility of an electroantennography study looking at the responses of parasitoids to the plant volatiles. Considering that the title suggests this is the most important finding from this work, leaving it incomplete for the publication seems odd. Omitting to discuss any known behavioural effects of the compounds of interest in this study is inexplicable. As it stands the results show a possible correlation, but not a functional relationship between these compounds and the insect responses. It is likely, but not certain, that *A. ervi* responds to one or more of these compounds.

The authors state that the parasitoid responses to plant volatiles are well studied, however, as far as I can see from looking at the references that have been added, none of the compounds which are induced and of interest in this manuscript are common with the compounds which are studied in the references the authors cite. I cannot find any references where *Aphidius ervi* is shown to respond to these compounds on their own or in mixtures.

Takemoto and Takabayashi (2015) identify a number of compounds in *Acyrtosiphum pisum* infested vs uninfested plants. A number of those compounds are the same as in Table S2 in this study, but they only tested behavioural responses to a small subset, three of which correspond with identifications in Table S2 (β -myrcene, E- β -ocimene and linalool) and of those there was no significant difference in quantities between symbiont and symbiont free treated plants. Of those only linalool shows a repellent effect on *A. ervi* at low concentrations, but in a blend it becomes attractive.

Guerrieri et al (1999) only test responses to air entrainment extracts and no chemical identification is performed.

Du et al (1998) again test a number of compounds which are listed in TableS2, but with no significant difference in quantity between treatments.

It is therefore puzzling how the identified compounds are involved in the behavioural changes of *A. ervi* in this study when compounds identified as attractants in other work performed by different research groups do not seem to be involved. Here the loss of attraction is related to reduced concentration of compounds which are not previously known to elicit attraction by *A. ervi*, and are not known to be induced by aphid feeding in similar studies, hence this needs to be elucidated.

Could there be a residual effect on the aphid saliva from the antibiotic treatment which is affecting the volatile production of the plant, explaining the differences observed in this study and previous work? It is not clear from the manuscript whether the *H. defensa* aphid clone was a natural infection or a

laboratory one and if so, had the original clone been treated with antibiotics? The fact that in experiments carried out with *R. insecticola* there was not a behavioural effect with aphids having natural symbiont associations, but an effect shown with an artificial introduction of the symbiont (lines 299-303), this needs to be clarified.

Coupled Gas-Chromatography-Electroantennography (GC-EAG) would show which compounds in the volatile profile of each sample the insects respond to, this could then be related back to the compounds identified in higher concentrations in the treatment (and others), as well as in the literature. Even without the bioassays to test for responses to those compounds, this would show whether these compounds and/or the different concentration present is of biological relevance to the parasitoid. As the sampling was done with Tenax I presume there are no spare liquid sample to use for GC-EAG, which complicates a follow on experiment, but if the entrainment samples are still available, GC-EAG would be very straightforward, to at least establish a functional link between the insect behaviour and the identified compounds. If not, going directly to behavioural bioassays with a range of concentration of the individual compounds and blends in the ratio found in the air entrainment samples would be the second-best option.

There is no discussion on what is known about the compounds of interest from this study. Are they known to elicit behavioural responses to other insect species? Are they known to be plant stress signals picked up by aphid/insect predators?

Line 142. Delete sentence "The attraction ... Takabayashi 2015)." I appreciate the authors expanding on this, but it is also referred to in the introduction.

Line 346. Before looking at the effect at a molecular level, the association between the insect behaviour and plant volatile chemistry needs to be confirmed.

Reviewer #3 (Remarks to the Author):

In the first paragraph of the results, I understand that you don't want to present all the statistical results because the statements are encompassing the results of multiple experiments. However, it is not clear to the reader that this is the case, and it would help to say something like, "across a large set of experiments with different..."

Also, in this same paragraph, when you provide the reference to the figure, indicate that statistical results can be found in the figure caption.

One result that stands out involves the *Regiella* results in Experiment 4. In your discussion, you argue that variation in whether plants fed on by symbiont-infected aphids are more attractive to parasitoids is not surprising given the variable effects of symbiont strains on many phenotypes. However, this does not explain why antibiotic clearance of *Regiella* 319 has no effect while microinjection of 319 has a pronounced effect. Are these two experiments performed in the same host genotype? Based on the caption to Table 1, I don't believe they are, but I had to search for a while to find this buried information. To make this more clear, the experimental details for experiment 4 need to be extended beyond their one sentence, and the figure caption for the experimental results should also address this difference in host background. In addition, in the discussion, your statement reads as if we expect variation based on symbiont genotype, but these data imply variation based on host background as well, and this needs to be clarified.

Finally, while reviewer 2's suggestion of additional experiments would add an additional layer of

verification, given the background work that has already been completed, I do not believe these additional experiments are necessary for publication.

signed,
Nicole Gerardo

Reviewer #4 (Remarks to the Author):

This is an interesting study, which shows that aphid symbionts can alter the induced volatile profiles of host plants and thereby influence the recruitment of aphid parasitoids. In general I think the data presented are convincing on this point. However, I think some care is warranted in interpreting the broader significance of these results.

It would certainly be interesting to know whether symbionts have a particularly pronounced (i.e., targeted) effect on specific aspects of the volatile blend that mediate wasp attraction. I tend to agree with the authors that exploring this issue through additional behavioral assays along with electrophysiology would be a complex undertaking, the scope of which would likely warrant a separate study. That said, I think the analysis and discussion of the volatile cues in the current paper is somewhat lacking. For one thing, it would be interesting to know whether the symbiont effects are focused on compounds that are induced by aphid feeding and thus likely provide informative cues for the wasps. Are the symbionts suppressing the aphid-induced volatile response, as opposed to just altering or reducing the overall blend? This seems like a basic point that isn't explicitly addressed by the current data. It does appear that the most pronounced effect of symbionts is to reduce volatile emissions. I didn't see a discussion of overall emission levels, but the 9 compounds highlighted by the authors as statistically significant were all lower in the presence of *Hamiltonella defensa*, and Table S2 suggests that there is a broad tendency for the symbiont to lower the emission of many compounds. Given this pattern, a likely explanation of the behavioral findings reported is that the presence of symbionts simply reduce the production of the relevant cues mediating parasitoid recruitment (notwithstanding the presence of statistically distinguishable qualitative differences between the blends). Perhaps the authors have a different view, but I think this possibility should be addressed.

Another way to investigate the specific cues mediating wasp behavior would be to look for similarity in the effects of different symbionts (which were found to have similar effects on wasp behavior) on volatile emissions. Do the other symbionts target the same compounds identified as being strongly influenced by *H. defensa*? Or is there just a shared tendency to reduce overall emissions, with different compounds showing up as most important for different symbionts? Providing volatile data for the other symbionts would also inform the author's speculation (discussed further below) that symbionts that do not confer parasitoid resistance in aphids might mimic the induced volatile cues of those that do.

Another interesting experiment would be to examine induced plant defense responses upstream of induced volatile emissions, for example by looking at defense phytohormone induction, to explore whether the observed effects are specifically targeted to volatiles. Perhaps the observed effects are instead just one aspect of a broader effect of symbionts that tends to attenuate the overall induction of plant defense responses? The authors do discuss the potential effects of symbiont-derived elicitors on plant responses and point to future work on molecular mechanisms. However, a basic characterization of plant responses to aphids with and without symbionts would complement the current data and help to clarify whether the observed effects reflect a targeted manipulation of plant volatiles or are potentially just one aspect of a broader effect on plant defense responses.

I don't think the additional experiments suggested above are completely essential for the current paper. If the significance of the current paper rests on showing a novel effect of aphid symbionts on volatile mediated interactions with aphid parasitoids, the current data are sufficient. However, the discussion should address the limitations of the current data set, as discussed further below.

I assume the authors initially focused on *H. defensa* on the reasonable hypothesis that its effects on volatile cues mediating wasp recruitment might reflect (and potentially act as an aposematic signal for) its effects on aphid resistance to parasitism. If they had then found that the observed effects on wasp behavior were limited to symbiont strains that confer resistance against parasitism, that would have been highly suggestive that the effect was part of an anti-parasitoid strategy by the symbiont. Instead, they found a broad tendency of symbionts to reduce wasp attraction, possibly due to the attenuation of volatile emissions. To me that suggests the possibility of a broader phenomenon that may or may not primarily be driven by effects on parasitoid recruitment. Yet, the paper leans pretty heavily toward interpreting the results as an adaptive symbiont strategy focused on disrupting parasitoid foraging, particularly in the second paragraph of the discussion, starting on line 325. I think a somewhat broader perspective could be useful, considering how little we know about either the mechanisms or the evolutionary ecology driving this system.

The most problematic statement in the paragraph mentioned above is probably the speculation about Batesian mimicry on lines 334-337. First, because it is just wildly speculative. Based on the current data it is difficult to say much at all about the adaptive significance of the observed effects, much less whether those effects are adaptive for some symbiont strains because they act as aposematic signals of conferred resistance to parasitism and for other strains because they effectively mimic those signals. Beyond that, I'm not sure that the basic logic holds in this case. As discussed above, the most obvious feature of the observed effects on plant signaling phenotypes is to reduce emissions. Thus it seems likely that the observed effect on wasp behavior might be explained by reduced salience of cues from plants harboring symbiont-infected aphids (due either to attenuation of the overall blend or of key blend components). But the paradigmatic examples of Batesian mimicry involve mimics of chemically defended species with conspicuous warning coloration. If you found a cryptic butterfly species that was also well defended chemically, you would not jump to the conclusion that the other, less well defended, cryptic species were Batesian mimics. And that seems like a closer analogy to the current findings than the paradigmatic examples.

Reply to Reviewer #1

The authors have made a sincere revision, and addressed all of the concerns that I had in a reasonable way. I agree with the authors that adding mechanistic antennography experiments would not add particularly useful information to the study.

Note that the changes made in the ms. were not highlighted in the text or listed by line number in the response to the reviewers so it took effort to evaluate if reasonable changes were made. One very minor question I still have- I had asked what a “population cage” is. The authors added to the text that “population cages are arenas large enough for parasitoids to show typical searching and oviposition behavior.” What size is that?

As already mentioned in the methods section, these cages were cubic gauze cages of 47.5 x 47.5 x 47.5 cm. Many behavioural experiments with aphid parasitoids using these cages have been done. We have added in the methods section a reference in which these same cages were used for behavioural experiments.

Reply to Reviewer #2

*I disagree with the authors in terms of the usefulness and feasibility of an electroantennography study looking at the responses of parasitoids to the plant volatiles. Considering that the title suggests this is the most important finding from this work, leaving it incomplete for the publication seems odd. Omitting to discuss any known behavioural effects of the compounds of interest in this study is inexplicable. As it stands the results show a possible correlation, but not a functional relationship between these compounds and the insect responses. It is likely, but not certain, that *A. ervi* responds to one or more of these compounds.*

*The authors state that the parasitoid responses to plant volatiles are well studied, however, as far as I can see from looking at the references that have been added, none of the compounds which are induced and of interest in this manuscript are common with the compounds which are studied in the references the authors cite. I cannot find any references where *Aphidius ervi* is shown to respond to these compounds on their own or in mixtures.*

*Takemoto and Takabayashi (2015) identify a number of compounds in *Acyrtosiphum pisum* infested vs uninfested plants. A number of those compounds are the same as in Table S2 in this study, but they only tested behavioural responses to a small subset, three of which correspond with identifications in Table S2 (β -myrcene, *E*- β -ocimene and linalool) and of those there was no significant difference in quantities between symbiont and symbiont free treated plants. Of those only linalool shows a repellent effect on *A. ervi* at low concentrations, but in a blend it becomes attractive.*

Guerrieri et al (1999) only test responses to air entrainment extracts and no chemical identification is performed.

Du et al (1998) again test a number of compounds which are listed in TableS2, but with no significant difference in quantity between treatments.

*It is therefore puzzling how the identified compounds are involved in the behavioural changes of *A. ervi* in this study when compounds identified as attractants in other work performed by different research groups do not seem to be involved. Here the loss of attraction is related to reduced concentration of compounds which are not previously known to elicit attraction by *A. ervi*, and are not known to be induced by aphid feeding in similar studies, hence this needs to be elucidated.*

*Could there be a residual effect on the aphid saliva from the antibiotic treatment which is affecting the volatile production of the plant, explaining the differences observed in this study and previous work? It is not clear from the manuscript whether the *H. defensa* aphid clone was a natural infection or a laboratory one and if so, had the original clone been treated with antibiotics? The fact that in experiments carried out with *R. insecticola* there was not a behavioural effect with aphids having natural symbiont associations, but an effect shown with an artificial introduction of the symbiont (lines 299-303), this needs to be clarified.*

Coupled Gas-Chromatography-Electroantennography (GC-EAG) would show which compounds in the volatile profile of each sample the insects respond to, this could then be related back to the compounds identified in higher concentrations in the treatment (and others), as well as in the literature. Even without the bioassays to test for responses to those compounds, this would show whether these compounds and/or the different concentration present is of biological relevance to the parasitoid. As the sampling was done with Tenax I presume there are no spare liquid sample to use for GC-EAG, which complicates a follow on experiment, but if the entrainment samples are still available, GC-EAG would be very straightforward, to at least establish a functional link between the insect behaviour and the identified compounds. If not, going directly to behavioural bioassays with a range of concentration of the individual compounds and blends in the ratio found in the air entrainment samples would be the second-best option.

We agree with the reviewer that the proposed experiments would be very interesting and would provide a functional characterisation of the different compounds found. We maintain, however, the main argument from our last revision as these experiments are far from the objectives of our project. Our volatile data provide clear evidence that volatile blends are

altered by the symbiont, and the behavioural experiments show that these changes are responsible for changes in wasp behaviour and ultimately aphid fitness. The experiment in which bean leaves upon which aphids fed were covered with aluminium foil shows that wasp responses are not due to any aphid residue in the leaf. An effect of the antibiotic on aphid saliva is also very unlikely. As we explain in the methods section, antibiotic treatments were performed at least 10 generations before the experiments.

We agree with this reviewer, however, that a discussion on the volatiles found to be in significantly lower amounts when the symbiont was present is needed. This point has also been raised by reviewer #4. Please see our reply to this reviewer, and the extra sentences added in the discussion section.

We also agree with this reviewer that the intriguing result with the symbiont *Regiella* needs further discussion. This point was also raised by reviewer #3, please see our comment to him/her.

There is no discussion on what is known about the compounds of interest from this study. Are they known to elicit behavioural responses to other insect species? Are they known to be plant stress signals picked up by aphid/insect predators?

We agree with the reviewer that compounds of interest need to be discussed. The potential role of the compounds β -cubebene and α -amorphen (also identified by Takemoto and Takabayashi (2015)) on wasp behaviour is now mentioned in the discussion.

Line 142. Delete sentence "The attraction ... Takabayashi 2015)." I appreciate the authors expanding on this, but it is also referred to in the introduction.

Done.

Line 346. Before looking at the effect at a molecular level, the association between the insect behaviour and plant volatile chemistry needs to be confirmed.

We prefer to maintain this sentence where we identify future avenues of research in this system. This includes the functional characterisation of the different volatiles found as well as the molecular details triggering the response.

Reply to Reviewer #3

In the first paragraph of the results, I understand that you don't want to present all the statistical results because the statements are encompassing the results of multiple experiments. However, it is not clear to the reader that this is the case, and it would help to say something like, "across a large set of experiments with different..."

Done.

Also, in this same paragraph, when you provide the reference to the figure, indicate that statistical results can be found in the figure caption.

We have included this information in the first mention of Figure 1.

One result that stands out involves the Regiella results in Experiment 4. In your discussion,

you argue that variation in whether plants fed on by symbiont-infected aphids are more attractive to parasitoids is not surprising given the variable effects of symbiont strains on many phenotypes. However, this does not explain why antibiotic clearance of Regiella 319 has no effect while microinjection of 319 has a pronounced effect. Are these two experiments performed in the same host genotype? Based on the caption to Table 1, I don't believe they are, but I had to search for a while to find this buried information. To make this more clear, the experimental details for experiment 4 need to be extended beyond their one sentence, and the figure caption for the experimental results should also address this difference in host background. In addition, in the discussion, your statement reads as if we expect variation based on symbiont genotype, but these data imply variation based on host background as well, and this needs to be clarified.

As shown in Table 1, the natural Regiella 319 was removed from an aphid clone collected on *Trifolium pratense*, and this same symbiont was injected into a clone obtained from *Lathyrus pratensis*. The effect of the symbiont was thus explored in two different host genotypes. We have clarified this in the methods section as well as in the figure legend. We have also added in the discussion section that the results observed may also depend on the interaction between the genotype of the symbiont and that of the aphid host.

Finally, while reviewer 2's suggestion of additional experiments would add an additional layer of verification, given the background work that has already been completed, I do not believe these additional experiments are necessary for publication.

*signed,
Nicole Gerardo*

We appreciate this remark, which we fully agree with.

Reply to Reviewer #4

This is an interesting study, which shows that aphid symbionts can alter the induced volatile profiles of host plants and thereby influence the recruitment of aphid parasitoids. In general I think the data presented are convincing on this point. However, I think some care is warranted in interpreting the broader significance of these results.

*It would certainly be interesting to know whether symbionts have a particularly pronounced (i.e., targeted) effect on specific aspects of the volatile blend that mediate wasp attraction. I tend to agree with the authors that exploring this issue through additional behavioral assays along with electrophysiology would be a complex undertaking, the scope of which would likely warrant a separate study. That said, I think the analysis and discussion of the volatile cues in the current paper is somewhat lacking. For one thing, it would be interesting to know whether the symbiont effects are focused on compounds that are induced by aphid feeding and thus likely provide informative cues for the wasps. Are the symbionts suppressing the aphid-induced volatile response, as opposed to just altering or reducing the overall blend? This seems like a basic point that isn't explicitly addressed by the current data. It does appear that the most pronounced effect of symbionts is to reduce volatile emissions. I didn't see a discussion of overall emission levels, but the 9 compounds highlighted by the authors as statistically significant were all lower in the presence of *Hamiltonella defensa*, and Table S2 suggests that there is a broad tendency for the symbiont to lower the emission of many compounds. Given this pattern, a likely explanation of the behavioral findings reported is*

that the presence of symbionts simply reduce the production of the relevant cues mediating parasitoid recruitment (notwithstanding the presence of statistically distinguishable qualitative differences between the blends). Perhaps the authors have a different view, but I think this possibility should be addressed.

We appreciate this remark, and although we believe wasp behaviour was mostly influenced by changes in a subset of volatile compounds, we acknowledge that overall volatile emissions may also play a role. To test this, and as suggested by this reviewer, we have performed an extra analysis to test the effect of the symbiont on total volatile emissions. This analysis reveals that the presence of the symbiont reduces total volatile emissions, and suggests that the defensive role of the symbionts may be more general than reducing parasitic wasp recruitment. We have added a new paragraph in the discussion where we explore this possibility, particularly on the potential role of overall volatile reductions in making plants less conspicuous to generalist aphid natural enemies like predators. As suggested by Reviewer #2, we have added a discussion on the potential role of two compounds (β -cubebene and α -amorphen) that in our study are emitted at significantly lower rates when the symbiont is present, and which were only emitted when plants were attacked by aphids in the study by Takemoto and Takabayashi (2015).

*Another way to investigate the specific cues mediating wasp behavior would be to look for similarity in the effects of different symbionts (which were found to have similar effects on wasp behavior) on volatile emissions. Do the other symbionts target the same compounds identified as being strongly influenced by *H. defensa*? Or is there just a shared tendency to reduce overall emissions, with different compounds showing up as most important for different symbionts? Providing volatile data for the other symbionts would also inform the author's speculation (discussed further below) that symbionts that do not confer parasitoid resistance in aphids might mimic the induced volatile cues of those that do.*

Another interesting experiment would be to examine induced plant defense responses upstream of induced volatile emissions, for example by looking at defense phytohormone induction, to explore whether the observed effects are specifically targeted to volatiles. Perhaps the observed effects are instead just one aspect of a broader effect of symbionts that tends to attenuate the overall induction of plant defense responses? The authors do discuss the potential effects of symbiont-derived elicitors on plant responses and point to future work on molecular mechanisms. However, a basic characterization of plant responses to aphids with and without symbionts would complement the current data and help to clarify whether the observed effects reflect a targeted manipulation of plant volatiles or are potentially just one aspect of a broader effect on plant defense responses.

I don't think the additional experiments suggested above are completely essential for the current paper. If the significance of the current paper rests on showing a novel effect of aphid symbionts on volatile mediated interactions with aphid parasitoids, the current data are sufficient. However, the discussion should address the limitations of the current data set, as discussed further below.

We agree with the reviewer that the experiments proposed would be very interesting and would provide additional support to our speculations, as well as a clearer mechanistic understanding on the patterns found. As he or she acknowledges, however, these studies would imply a remarkable extra effort that is far from the objectives of our study. Moreover, the additional experiments would likely raise additional mechanistic questions and consequently represent an additional project in itself. We have added a new paragraph in the

manuscript where we identify the limitations of our study in this regard. As explained in our previous response, and given the overall reduction in volatiles found in plants infested with aphids carrying the symbiont, we have also added a new paragraph discussing that the effect found may be a general defensive mechanism against generalist aphid natural enemies.

I assume the authors initially focused on H. defensa on the reasonable hypothesis that its effects on volatile cues mediating wasp recruitment might reflect (and potentially act as an aposematic signal for) its effects on aphid resistance to parasitism. If they had then found that the observed effects on wasp behavior were limited to symbiont strains that confer resistance against parasitism, that would have been highly suggestive that the effect was part of an anti-parasitoid strategy by the symbiont. Instead, they found a broad tendency of symbionts to reduce wasp attraction, possibly due to the attenuation of volatile emissions. To me that suggests the possibility of a broader phenomenon that may or may not primarily be driven by effects on parasitoid recruitment. Yet, the paper leans pretty heavily toward interpreting the results as an adaptive symbiont strategy focused on disrupting parasitoid foraging, particularly in the second paragraph of the discussion, starting on line 325. I think a somewhat broader perspective could be useful, considering how little we know about either the mechanisms or the evolutionary ecology driving this system.

This is a very interesting remark, and as explained in our previous response, we have added a new paragraph in the discussion section to accommodate this reviewer's suggestion.

The most problematic statement in the paragraph mentioned above is probably the speculation about Batesian mimicry on lines 334-337. First, because it is just wildly speculative. Based on the current data it is difficult to say much at all about the adaptive significance of the observed effects, much less whether those effects are adaptive for some symbiont strains because they act as aposematic signals of conferred resistance to parasitism and for other strains because they effectively mimic those signals. Beyond that, I'm not sure that the basic logic holds in this case. As discussed above, the most obvious feature of the observed effects on plant signaling phenotypes is to reduce emissions. Thus it seems likely that the observed effect on wasp behavior might be explained by reduced salience of cues from plants harboring symbiont-infected aphids (due either to attenuation of the overall blend or of key blend components). But the paradigmatic examples of Batesian mimicry involve mimics of chemically defended species with conspicuous warning coloration. If you found a cryptic butterfly species that was also well defended chemically, you would not jump to the conclusion that the other, less well defended, cryptic species were Batesian mimics. And that seems like a closer analogy to the current findings than the paradigmatic examples.

We have removed this speculation from the text and replaced it by the hypothesis that the effect found may be a more general defensive strategy aimed at avoiding attacks by generalist predators (see comments above).

REVIEWERS' COMMENTS:

Reviewer #4 (Remarks to the Author):

My most serious concern about the previous submission was that the authors too aggressively interpreted their findings in light of the initial hypothesis that the effects of *H. defensa* on volatile cues mediating wasp recruitment might complement its effects on aphid resistance to parasitism, neglecting the strong possibility that the results obtained might reflect broader effects of *H. defensa*, as well as other symbionts (including those that do not confer resistance), on host-plant defenses in general or other aspects of plant physiology.

The revised discussion has largely addressed my concerns, although, I will comment on one point: In their response to my previous review, the authors wrote, "...although we believe wasp behaviour was mostly influenced by changes in a subset of volatile compounds, we acknowledge that overall volatile emissions may also play a role." and in the revised text they write, "a possible interpretation of these results is that presence of symbionts in aphids generally makes plants less conspicuous to all aphid natural enemies that use quantitative, rather than qualitative, information in plant volatile plumes." In each case, the authors seem to pose qualitative and quantitative effects as competing hypotheses, and I'm not sure that this is appropriate. I agree that the behavior of the wasp is probably dependent on only a subset of the compounds. The point of my previous remarks was that an overall attenuation of volatiles would also likely to reduce the salience of whatever specific cues the wasps use (even if you are focussed on qualitative information, quantitative effects can still be important).

Beyond that, I think the revised sections of the discussion, while improved with respect to content, are slightly less polished than the rest of the text with respect to writing (probably just because they have been through fewer drafts). To give one example, on line 331 I think ",including" would probably convey the intended meaning better than "as well as". In general I would suggest that the authors give the discussion another look prior to publication.

Reply to Reviewer #4

*My most serious concern about the previous submission was that the authors too aggressively interpreted their findings in light of the initial hypothesis that the effects of *H. defensa* on volatile cues mediating wasp recruitment might complement its effects on aphid resistance to parasitism, neglecting the strong possibility that the results obtained might reflect broader effects of *H. defensa*, as well as other symbionts (including those that do not confer resistance), on host-plant defenses in general or other aspects of plant physiology.*

The revised discussion has largely addressed my concerns, although, I will comment on one point: In their response to my previous review, the authors wrote, "...although we believe wasp behaviour was mostly influenced by changes in a subset of volatile compounds, we acknowledge that overall volatile emissions may also play a role." and in the revised text they write, "a possible interpretation of these results is that presence of symbionts in aphids generally makes plants less conspicuous to all aphid natural enemies that use quantitative, rather than qualitative, information in plant volatile plumes." In each case, the authors seem to pose qualitative and quantitative effects as competing hypotheses, and I'm not sure that this is appropriate. I agree that the behavior of the wasp is probably dependent on only a subset of the compounds. The point of my previous remarks was that an overall attenuation of volatiles would also likely to reduce the salience of whatever specific cues the wasps use (even if you are focussed on qualitative information, quantitative effects can still be important).

We have fully rewritten this paragraph, and have added that: "We also found that total volatile emissions were significantly reduced by the presence of *H. defensa*. A possible interpretation of this result is that aphid symbionts reduce parasitic wasp recruitment by suppressing signalling pathways downstream of the production of multiple volatile compounds".

Beyond that, I think the revised sections of the discussion, while improved with respect to content, are slightly less polished than the rest of the text with respect to writing (probably just because they have been through fewer drafts). To give one example, on line 331 I think ",including" would probably convey the intended meaning better than "as well as". In general I would suggest that the authors give the discussion another look prior to publication.

We have carefully read the whole manuscript and edited when necessary.